# Understanding displacement of onboard contingents in Navy amphibious ships

Heitor Martinez-Grueira[1], Rafael Asorey-Cacheda[2]*, Antonio-Javier Garcia-Sanchez[2], Joan Garcia-Haro[2]

1 Centro Militar de Farmacia de la Defensa, Ministry of Defense, Colmenar Viejo, Madrid, Spain,
2 Department of Information and Communications Technologies, Universidad Politécnica de Cartagena, Cartagena, Región de Murcia, Spain

☺ All these authors are contributed equally to this work.
* rafael.asorey@upct.es

**Data Availability Statement:** All relevant data are within the paper.

**Funding:** This work was a result of the ThinkInAzul and AgroAINext programmes, funded by Ministerio

## Abstract

The Naval Ship Code (NSC) was enacted in 2009 to standardize regulations for NATO member naval forces, and a study commissioned by the Spanish Navy General Staff (EMA) aimed to identify the factors that influence onboard personnel's ability to move during an evacuation process. This study validated the soundness of the safety protocols implemented on navy vessels and highlighted the impact of certain characteristics of the embarked military contingent, such as body mass index, age, and seniority. It also found that such characteristics could act as distinctive factors among the embarked contingents in the evacuation of a military vessel. The study quantified the effect of these intervening characteristics, confirming the need for different displacement models for each of the study contingents to improve ship evacuation maneuvers. The findings of this study provide insights into the behavior of different embarked contingents during the evacuation process and can inform the development of more effective safety protocols for military naval operations. The starting hypothesis is that certain characteristics of the embarked military contingent have a decisive influence on their displacement capacity during the evacuation process. This hypothesis has been expanded in the sense that these same characteristics can act as differentiating elements among the embarked contingents evacuating a military vessel. It is possible to quantify the influence of these characteristics and implement a displacement model applicable in escape, evacuation, and rescue processes. Thus, the specific characteristics of a study contingent will be reflected in its displacement model. In this article we find that while members of the landing force (LF) show greater displacement capacity through a longitudinal corridor (around 10%), their ability to overcome other passage elements present on the study vessel is reduced (around 30%) compared to members of the vessel's own crew.

## Introduction

Passenger and crew safety on civilian ships lacked proper technical or legislative consideration until well into the 20th century [1–3]. On the other hand, this topic has always been present in

de Ciencia, Innovación y Universidades (MICIU) with funding from European Union NextGenerationEU/PRTR-C17.I1 and by Fundación Séneca with funding from Comunidad Autónoma Región de Murcia (CARM). This work was also supported by the grants PID2023-148214OB-C21 and TED2021-129336B-I00, funded by MICIU/AEI/ 10.13039/501100011033 and by the European Union NextGenerationEU/PRTR. This work was also funded by Fundación Séneca (22236/PDC/23). This research was also contextualized to DAIMon, a cascade funding action deriving from the Horizon Europe project aerOS, funded by the European Commission under grant number 101069732. The funders had no role in study design, data collection and analysis, decision to publish, or preparation of the manuscript.

**Competing interests:** The authors have declared that no competing interests exist.

the military environment due to its highly-qualified personnel, whose duties can involve risks to the ship and themselves. This singularity has put the Navy at the forefront of technical and technological innovation in the field of maritime safety.

The change in the Spanish Armed Forces (FAS) model at the beginning of this century caused naval safety to take a back seat until the NSC and its chapter VII dealing with Escape, Evacuation, and Rescue (EER) was issued. This subject is strengthened in the current version of the NSC [4], although it is still general and, in many cases, incompatible with the military environment as it collects data and references from civil regulation standards in the framework of the International Maritime Organization (IMO) [5, 6].

This is one of the reasons justifying the study carried out by the Spanish Military Operational Research Cabinet (GIMO), given the scarce or nonexistent application of personnel distribution or the standards in the NSC for military vessels based again on the IMO regulations for civilian ships. Among its particularities, it is worth mentioning the unrealistic ratios of gender and age for a military vessel, as well as its reference to passengers with reduced mobility or the distinction between passengers and crew [7].

Thus, it was necessary to undertake a rigorous study that would include a preliminary phase, during which a range of tests would be executed to gather essential data (affiliation, anthropometric, professional, behavioral, etc.) about the Navy's personnel. The data collected and the conclusions of the study would consolidate a knowledge base from which to deploy new, more specific, and ambitious research lines [8].

This paper follows the logical assumption that different contingents present different displacement models, which implies acquiring new data once the obsolescence of the existing data had been confirmed. Thus, the same range of tests designed in the original study was executed by GIMO, focusing on the most common obstacles or passageways of a military vessel. This approach enriched the research with data obtained aboard an amphibious vessel using a Statistical Design of Experiments (SDE) [9], allowing for the sizing of sample contingents composed of the vessel's crew (VC) and a landing force of Marine Infantry personnel.

By using the SDE methodology, it was possible to identify the population's distinctive features (body mass index –BMI–, age, and seniority) that truly play a role in the study process and, at the same time, to determine the extent of that role. Then, a displacement model for each of the reference contingents was developed.

An ANOVA analysis was performed on the data obtained from the study contingents, where the null hypothesis of equal means was rejected, leading to the conclusion that there was not enough data to assume that both contingents belong to the same population. Thus, an independent displacement model for each contingent is reasonably justified. The model consists of functions that quantify the displacement capacity (time) of an individual belonging to a specific contingent based on their particular characteristics and the different obstacles on the military vessel studied during the evacuation process [10].

The rest of this paper is organized as follows. Section "Material and methods" describes the material and methods. Section "Data collection" presents the data collection process. Section "Dataset integration" shows how the dataset was integrated. Section "Comparing tests between populations" compares and analyzes the results achieved for each of the populations. In Section "Summary of the results", a summary of the results is presented and discussed. Finally, Section "Conclusions" provides the closing remarks.

## Material and methods

To begin the new research, an SDE was planned at an early stage [11] to determine the size of the necessary population sample and identify the characteristics that affect their ability to

move during evacuation from the naval platform. We concluded that BMI, age, and seniority [12] decisively influence the movement of a contingent through the passageway elements tested [13, 14].

The current Military Personnel Law, enacted in 2007, foresaw the gradual aging of members of the Spanish Armed Forces who maintain a temporary professional relationship. Since age has a significant impact on the movement model of a contingent during an evacuation process, the data from previous investigations could be outdated.

Due to this risk, collecting updated data is necessary to determine whether the previously identified characteristics continue to influence the deployment of military personnel on board and to what extent (see example of Fig 1). To achieve this, the same range of tests planned in the original study would be executed to gain the necessary insight and implement a new displacement model for the reference personnel if necessary.

To conduct the range of tests, the Spanish Navy was asked to designate the amphibious assault ship "Galicia" (L-51) as the study vessel during the course of regular national maneuvers in the Gulf of Cadiz and all participating military personnel provided informed consent. This type of ship was used since the platform's and its personnel's characteristics needed to match the planned research. This would be challenging due to the vessel's dimensions, the size of the reference population, and the heterogeneity of the personnel on board.

This type of ship performs very diverse functions requiring specific personnel to provide the necessary capabilities (landing force, flight units, medical personnel, staff, etc.). Furthermore, the deployment of this personnel serves a specific and temporary function that depends on the duration of their operations. This temporality limits the knowledge of the general layout of the ship and training and instruction in the vessel's emergency procedures.

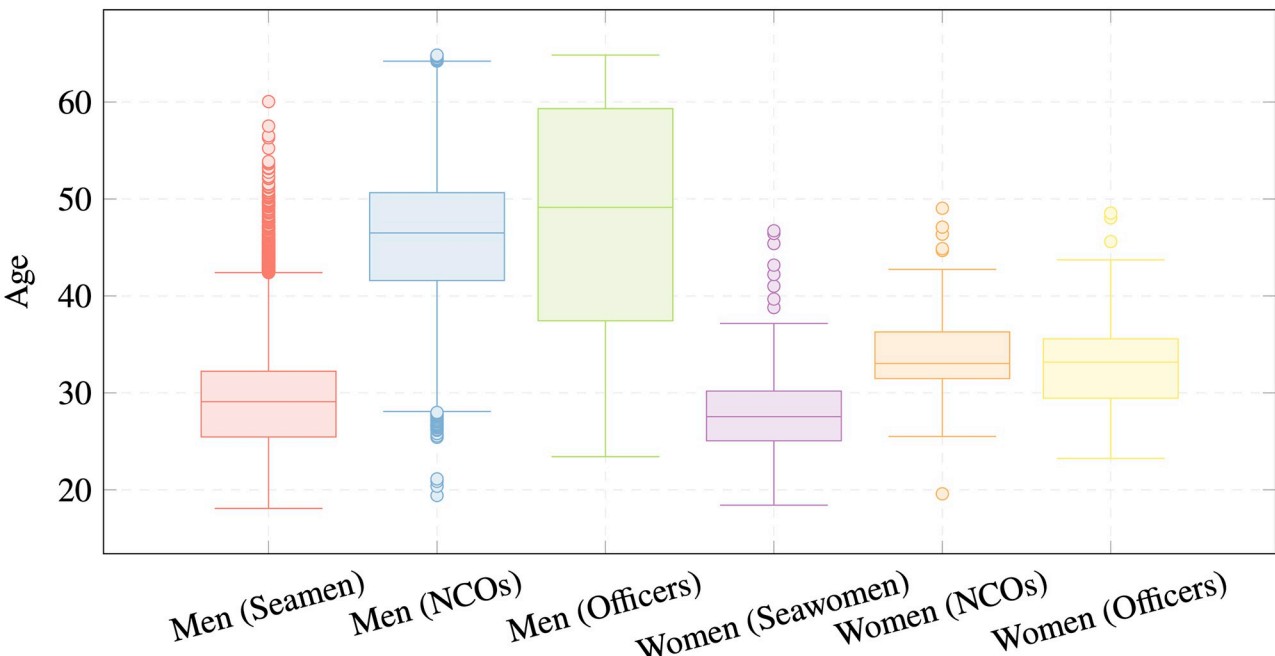

**Fig 1. Staff age by category and gender.** Spanish Navy staff age by category and gender in 2017 [15].

Among the personnel on board this type of ship, the landing force (LF), composed of members of the Marine Corps, stands out for its size, which sometimes doubles the size of the VC. Moreover, its temporality is limited to the duration of the planned amphibious operation, after which this force will leave the vessel. This temporality greatly restricts the knowledge this force has of the general layout of the ship and its emergency response protocols, compared to the VC's level of training and instruction.

However, considering the LF as mere passengers would be an error, as it is a military contingent and, therefore, has basic instruction and training in the vessel's emergency plans and procedures. In accordance with the EMA's regulations [16] for this type of ship, the embarked contingents are mandatory participants in periodically scheduled training programs, with the objective of instructing them in knowledge of the vessel and its procedures, including the EER plan.

## Data collection

Drawing from extensive experience in data collection and analysis, a comprehensive set of tests was designed, taking into account prior investigations [17, 18]. Using the Statistical Design of Experiments (SDE), we determined the necessary sample size, thus eliminating the necessity to involve the entire population on board, particularly during active maneuvers. The SDE methodology allows for the formation of control groups (population samples) based on members' specific characteristics. As a result, the influence of these characteristics on the study process is quantified, ensuring a representative study without the need for more resource-intensive methods, such as stratified random sampling [9].

To carry out an SDE, it is necessary to know the study population and its characteristics. With this information, the control groups that will participate in the tests to collect data (time measurements) are formed to make the test representative. This technique also allows us to determine the degree of involvement of each of the study factors and, thus, conclude which factors should be discarded if they have little or no influence.

During the tests, the time taken by each member of the control groups to pass through the following common passageways or overcome obstacles along the evacuation routes of the reference platform was recorded:

1. Movement test in a 1 m wide, 5 m long corridor (Fig 2): Each member of the control groups moved through the hallway at a swift pace, but without running.

2. Obstacle test, which involves ascending a ladder to the upper deck via a watertight hatch that is closed (Fig 3): The time taken by each member of the control groups to access, open, and overcome this obstacle, and then close the watertight hatch was recorded.

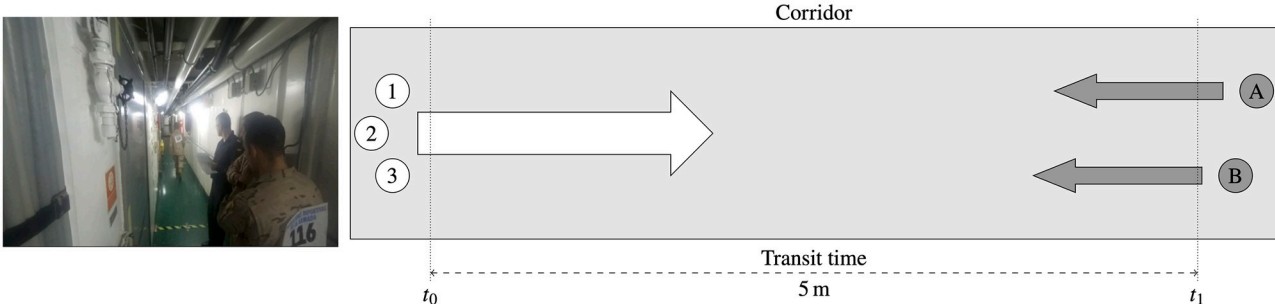

**Fig 2. First experiment.** Picture and diagram of the machine room hallway (first experiment).

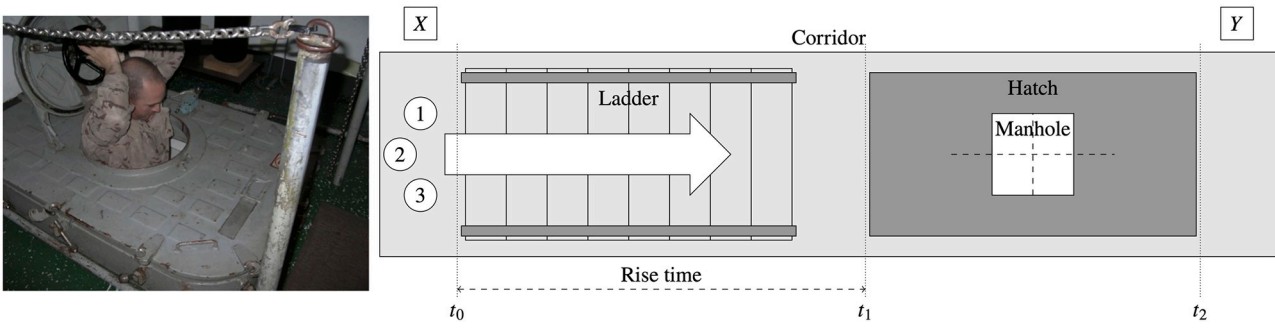

**Fig 3. Second experiment.** Picture and diagram of ladder and hatch with manhole (second experiment).

3. Door-clearing experiment (Fig 4): The time taken by each member of the control groups to open, pass through, and close the watertight door located in the main corridor was recorded.

All these tests were conducted with normal lighting to determine the time taken by each individual under specific environmental conditions during their execution.

## Dataset integration

After collecting the measurements from the tests carried out by members of the sample populations (LF and VC) on board the reference vessel, sufficient data was provided to build the SDE project, generating a displacement model for each test. These data also went through a previous phase of data filtering and analysis.

Once the collected data were validated, the SDE project involved constructing a matrix of the multifactorial experimental plan to reference the experiments that make up each study test (corridor displacement, opening and closing of watertight doors, going through a hatch between decks) by combining the significance levels of their study factors (BMI, age, seniority).

As a preliminary step to the construction of the experimental plan matrix, it was necessary to build a table (see Table 1), where each experiment is identified by an inverted binary system that combines the assigned significance levels (high = 1 or low = −1) for each study factor (A

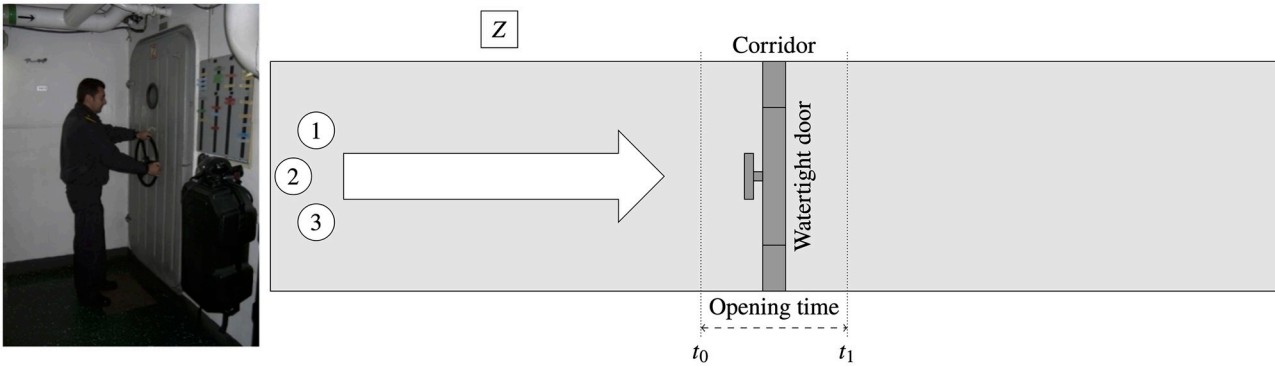

**Fig 4. Third experiment.** Picture and diagram of the escape trunk of the machinery room (third experiment).

**Table 1. Experimental design matrix.**

| ID | Experiment plan matrix | | | | | | |
|---|---|---|---|---|---|---|---|
| | Factors | | | Interactions | | | |
| | **A** | **B** | **C** | **AB** | **AC** | **BC** | **ABC** |
| 0 | -1 | -1 | -1 | 1 | 1 | 1 | -1 |
| 1 | 1 | -1 | -1 | -1 | -1 | 1 | 1 |
| 2 | -1 | 1 | -1 | -1 | 1 | -1 | 1 |
| 3 | 1 | 1 | -1 | 1 | -1 | -1 | -1 |
| 4 | -1 | -1 | 1 | 1 | -1 | -1 | 1 |
| 5 | 1 | -1 | 1 | -1 | 1 | -1 | -1 |
| 6 | -1 | 1 | 1 | -1 | -1 | 1 | -1 |
| 7 | 1 | 1 | 1 | 1 | 1 | 1 | 1 |

for BMI, B for age, C for seniority). This combination determined the characteristics of the experiments and, therefore, the members who made up the control group (sample population) participating in each test.

The levels of significance (high, low) assigned to each study factor (BMI, age, and seniority) were defined by the mean value of each attribute reported by the sample population.

Based on the combination of factors and degree of significance, each experiment was provided with the data collected from the control group whose members verified the defined characteristics (example experiment factors: $A = 1$, $B = -1$, $C = 1$; individuals with higher BMI, lower age, and more seniority than their respective means). Given the size of the participating sample population, it was determined that each control group would be composed of a total of 10 individuals.

Therefore, since each test had eight experiments and given the control groups were made up of 10 people, each response matrix (data matrix) consisted of eighty measurements (population sample).

## Members of the LF for the study

On the basis of the data provided by the members of this sample population, the means attained for their study factors (BMI, age, and seniority) determined the levels of significance (see Table 2). These characteristics indicated an individual being assigned to a particular control group depending on the value of their attributes with respect to the factors (BMI, age, and seniority).

Thus, the measurements obtained for the individuals of each control group of the different experiments in each trial constituted the SDE's response matrix. This technique allowed us to

**Table 2. Factors and degree of significance for LF.**

| Complete multifactorial trial plan | | |
|---|---|---|
| **Factor** | **Level** | **Value** |
| $A$—BMI | 1 | $> 24.33 \text{ kg/m}^2$ |
| | -1 | $< 24.33 \text{ kg/m}^2$ |
| $B$—Age | 1 | $> 26.74 \text{ yr}$ |
| | -1 | $< 26.74 \text{ yr}$ |
| $C$—Seniority | 1 | $> 4.44 \text{ yr}$ |
| | -1 | $< 4.44 \text{ yr}$ |

**Table 3. Response matrices, measurements from SDE trials (corridor displacement).**

| ID | Measurements | | | | | | | | | | | |
|----|------|------|------|------|------|------|------|------|------|------|------|------|
|    | $y_0$ | $y_1$ | $y_2$ | $y_3$ | $y_4$ | $y_5$ | $y_6$ | $y_7$ | $y_8$ | $y_9$ | $y_{av}$ | $y_{dev}$ |
| 0 | 3.11 | 3.14 | 2.81 | 3.48 | 3.51 | 2.92 | 3.49 | 3.22 | 3.70 | 3.52 | 3.29 | 0.29 |
| 1 | 2.06 | 2.60 | 2.69 | 2.65 | 2.89 | 2.66 | 2.84 | 2.78 | 3.06 | 3.83 | 2.77 | 0.44 |
| 2 | 3.06 | 3.33 | 3.12 | 3.11 | 3.70 | 3.20 | 2.98 | 2.82 | 3.42 | 3.30 | 3.20 | 0.19 |
| 3 | 2.74 | 2.90 | 3.25 | 2.80 | 2.69 | 2.69 | 2.83 | 2.79 | 2.76 | 2.72 | 2.77 | 0.21 |
| 4 | 3.53 | 3.08 | 2.78 | 3.26 | 3.83 | 3.66 | 3.60 | 3.12 | 3.90 | 2.75 | 3.29 | 0.32 |
| 5 | 2.62 | 3.21 | 3.69 | 3.56 | 3.34 | 2.97 | 3.24 | 3.84 | 3.46 | 3.16 | 3.19 | 0.34 |
| 6 | 3.11 | 2.79 | 3.45 | 3.24 | 2.74 | 3.20 | 3.81 | 3.23 | 3.23 | 3.09 | 3.19 | 0.34 |
| 7 | 2.62 | 3.21 | 3.69 | 3.56 | 3.34 | 2.97 | 3.24 | 3.84 | 3.46 | 3.16 | 3.31 | 0.36 |

**Table 4. Response matrices, measurements from SDE trials (opening and closing watertight doors).**

| ID | Measurements | | | | | | | | | | | |
|----|------|------|------|------|------|------|------|------|------|------|------|------|
|    | $y_0$ | $y_1$ | $y_2$ | $y_3$ | $y_4$ | $y_5$ | $y_6$ | $y_7$ | $y_8$ | $y_9$ | $y_{av}$ | $y_{dev}$ |
| 0 | 10.42 | 10.00 | 9.39 | 10.51 | 10.51 | 9.10 | 9.13 | 11.51 | 10.67 | 10.30 | 10.15 | 0.76 |
| 1 | 10.40 | 10.42 | 7.99 | 9.44 | 9.24 | 8.34 | 9.18 | 9.27 | 8.89 | 10.05 | 9.32 | 0.81 |
| 2 | 9.74 | 8.68 | 8.99 | 9.43 | 9.34 | 8.48 | 9.12 | 9.66 | 8.74 | 9.03 | 9.12 | 0.75 |
| 3 | 10.49 | 8.11 | 7.99 | 12.87 | 10.87 | 11.89 | 10.80 | 9.77 | 11.29 | 7.59 | 10.17 | 1.98 |
| 4 | 8.52 | 13.00 | 14.41 | 9.28 | 13.58 | 12.60 | 12.80 | 8.73 | 5.55 | 13.73 | 10.82 | 3.17 |
| 5 | 10.42 | 10.00 | 9.39 | 10.51 | 10.51 | 9.10 | 9.13 | 11.51 | 10.67 | 10.30 | 10.15 | 0.76 |
| 6 | 7.47 | 8.34 | 17.16 | 10.31 | 11.33 | 11.52 | 12.75 | 7.16 | 11.01 | 11.26 | 10.13 | 3.29 |
| 7 | 8.15 | 7.49 | 9.38 | 9.60 | 10.50 | 8.90 | 9.00 | 10.00 | 8.09 | 8.19 | 8.93 | 0.95 |

quantify the basic contributions and effects of the study factors to obtain the regression function that models the displacement capacity [19] for an LF subject based on their individual attributes (see Tables 3–5).

## Members of the VC for the study

Based on the data provided by the members of this sample population, the averages obtained for their study factors (BMI, age, and seniority) determined the significance levels (see Table 6). These characteristics also established whether an individual was assigned to one

**Table 5. Response matrices, measurements from SDE trials (hatch crossing between decks).**

| ID | Measurements | | | | | | | | | | | |
|----|------|------|------|------|------|------|------|------|------|------|------|------|
|    | $y_0$ | $y_1$ | $y_2$ | $y_3$ | $y_4$ | $y_5$ | $y_6$ | $y_7$ | $y_8$ | $y_9$ | $y_{av}$ | $y_{dev}$ |
| 0 | 30.66 | 26.16 | 32.45 | 35.00 | 23.49 | 23.75 | 26.36 | 20.31 | 28.39 | 23.14 | 26.97 | 4.63 |
| 1 | 20.74 | 24.52 | 35.04 | 29.25 | 35.51 | 26.16 | 28.52 | 28.01 | 30.25 | 25.53 | 27.52 | 4.57 |
| 2 | 27.83 | 27.60 | 28.14 | 28.03 | 27.72 | 27.55 | 27.46 | 27.78 | 27.48 | 27.60 | 27.72 | 0.16 |
| 3 | 26.34 | 37.00 | 28.60 | 18.84 | 23.71 | 27.09 | 24.26 | 28.87 | 25.61 | 23.97 | 25.25 | 6.01 |
| 4 | 21.74 | 30.37 | 19.17 | 23.30 | 28.16 | 31.14 | 20.78 | 24.47 | 38.03 | 17.24 | 24.89 | 6.10 |
| 5 | 20.74 | 24.52 | 35.04 | 29.25 | 35.51 | 26.16 | 28.52 | 28.01 | 30.25 | 25.23 | 27.52 | 4.57 |
| 6 | 18.83 | 22.60 | 22.69 | 29.95 | 19.91 | 28.53 | 19.63 | 24.94 | 26.32 | 26.03 | 23.94 | 4.42 |
| 7 | 24.29 | 22.73 | 23.81 | 28.95 | 25.91 | 19.63 | 27.58 | 28.56 | 19.69 | 17.39 | 24.57 | 3.50 |

**Table 6. Factors and degree of significance for VC.**

| Complete multifactorial trial plan | | |
|---|---|---|
| Factor | Level | Value |
| A—BMI | 1 | $> 25.70 \text{ kg/m}^2$ |
| | -1 | $< 25.70 \text{ kg/m}^2$ |
| B—Age | 1 | $> 35.31 \text{ yr}$ |
| | -1 | $< 35.31 \text{ yr}$ |
| C—Seniority | 1 | $> 12.26 \text{ yr}$ |
| | -1 | $< 12.26 \text{ yr}$ |

**Table 7. Response matrices, measurements from SDE trials (corridor displacement).**

| ID | Measurements | | | | | | | | | | | |
|---|---|---|---|---|---|---|---|---|---|---|---|---|
| | $y_0$ | $y_1$ | $y_2$ | $y_3$ | $y_4$ | $y_5$ | $y_6$ | $y_7$ | $y_8$ | $y_9$ | $y_{av}$ | $y_{dev}$ |
| 0 | 3.21 | 2.77 | 3.51 | 3.24 | 2.74 | 3.25 | 3.66 | 3.33 | 3.52 | 2.65 | 3.19 | 0.35 |
| 1 | 2.99 | 3.53 | 3.88 | 2.66 | 3.24 | 3.41 | 3.37 | 2.84 | 2.72 | 2.91 | 3.24 | 0.40 |
| 2 | 3.61 | 2.84 | 2.05 | 3.55 | 3.57 | 3.23 | 2.98 | 3.17 | 3.05 | 1.18 | 3.14 | 0.61 |
| 3 | 4.64 | 4.29 | 4.51 | 4.75 | 4.26 | 4.11 | 4.47 | 4.48 | 4.53 | 4.21 | 4.47 | 0.25 |
| 4 | 3.95 | 3.63 | 3.45 | 3.19 | 3.45 | 3.62 | 3.60 | 3.92 | 3.26 | 3.43 | 3.68 | 0.25 |
| 5 | 3.63 | 2.99 | 3.31 | 3.34 | 3.13 | 3.20 | 3.48 | 2.83 | 2.69 | 2.82 | 3.31 | 0.32 |
| 6 | 2.55 | 2.67 | 3.95 | 3.88 | 3.02 | 3.68 | 2.93 | 6.96 | 3.39 | 3.21 | 3.62 | 1.27 |
| 7 | 2.82 | 3.45 | 3.95 | 3.29 | 3.05 | 3.49 | 3.83 | 3.53 | 2.57 | 2.57 | 3.26 | 0.49 |

control group or another, depending on the value of their attributes with respect to the factors (BMI, age, and seniority).

To achieve this, the measurements collected from the individuals in each control group of the various experiments conducted in each trial constituted the values in the response matrix of the SDE. Using this method, the basic contributions and effects of the study factors were quantified to obtain a regression function that models the displacement capacity [14] of a member of the crew based on their individual characteristics (see Tables 7–9).

The application of SDE allowed us to form control groups (population samples) that were representative of the original population and its inherent characteristics. These groups executed the planned tests to gather sufficient data to develop a displacement model. The captured data was then subjected to various tests [20, 21], refinement processes (Chauvenet's criterion

**Table 8. Response matrices, measurements from SDE trials (opening and closing of watertight doors).**

| ID | Measurements | | | | | | | | | | | |
|---|---|---|---|---|---|---|---|---|---|---|---|---|
| | $y_0$ | $y_1$ | $y_2$ | $y_3$ | $y_4$ | $y_5$ | $y_6$ | $y_7$ | $y_8$ | $y_9$ | $y_{av}$ | $y_{dev}$ |
| 0 | 8.00 | 6.81 | 5.41 | 7.35 | 6.41 | 4.74 | 6.46 | 6.29 | 6.20 | 6.17 | 6.38 | 0.91 |
| 1 | 7.20 | 8.08 | 6.23 | 6.00 | 5.91 | 7.42 | 6.56 | 6.46 | 5.89 | 7.13 | 6.73 | 0.76 |
| 2 | 8.97 | 5.61 | 5.44 | 6.77 | 6.08 | 8.56 | 9.26 | 7.55 | 7.32 | 7.04 | 6.95 | 1.65 |
| 3 | 6.60 | 8.06 | 7.98 | 7.70 | 6.84 | 7.90 | 9.19 | 6.85 | 6.15 | 8.87 | 7.33 | 1.03 |
| 4 | 6.70 | 7.18 | 6.19 | 6.30 | 7.06 | 6.23 | 7.23 | 6.98 | 6.54 | 6.67 | 6.69 | 0.50 |
| 5 | 9.09 | 5.32 | 5.77 | 4.39 | 8.53 | 5.68 | 5.12 | 7.78 | 9.08 | 5.56 | 6.73 | 2.06 |
| 6 | 7.06 | 6.46 | 7.52 | 6.09 | 5.54 | 6.03 | 6.89 | 5.60 | 4.98 | 8.31 | 6.45 | 1.01 |
| 7 | 5.82 | 5.74 | 6.53 | 5.30 | 7.45 | 5.65 | 6.84 | 6.82 | 3.56 | 5.56 | 5.93 | 1.08 |

**Table 9. Response matrices, measurements from SDE trials (hatch crossing between decks).**

| ID | Measurements | | | | | | | | | | | |
|----|------|------|------|------|------|------|------|------|------|------|----------|-----------|
| | $y_0$ | $y_1$ | $y_2$ | $y_3$ | $y_4$ | $y_5$ | $y_6$ | $y_7$ | $y_8$ | $y_9$ | $y_{av}$ | $y_{dev}$ |
| 0 | 18.61 | 14.70 | 15.48 | 16.84 | 17.68 | 11.74 | 14.62 | 15.02 | 23.52 | 17.58 | 16.58 | 3.14 |
| 1 | 17.27 | 21.24 | 14.05 | 13.85 | 18.75 | 15.63 | 17.38 | 21.10 | 17.77 | 18.39 | 16.58 | 3.14 |
| 2 | 20.95 | 14.40 | 13.49 | 11.48 | 15.99 | 19.03 | 17.02 | 15.82 | 16.63 | 15.41 | 16.05 | 3.27 |
| 3 | 13.49 | 13.06 | 13.24 | 14.60 | 13.41 | 13.68 | 13.11 | 13.28 | 14.05 | 15.01 | 13.60 | 0.69 |
| 4 | 18.01 | 16.35 | 14.27 | 21.24 | 25.36 | 18.48 | 16.73 | 20.59 | 13.21 | 14.27 | 17.85 | 3.75 |
| 5 | 19.91 | 12.31 | 11.732 | 18.35 | 19.91 | 14.99 | 13.40 | 11.90 | 10.95 | 25.94 | 15.47 | 4.29 |
| 6 | 18.01 | 16.35 | 14.27 | 21.24 | 25.36 | 18.48 | 16.73 | 20.59 | 13.21 | 14.27 | 17.85 | 2.82 |
| 7 | 15.03 | 13.16 | 12.88 | 11.02 | 13.87 | 16.03 | 19.15 | 16.99 | 16.00 | 16.00 | 15.01 | 2.34 |

[22]), and a regression analysis based on the Shieffle equation to create the displacement function for a reference contingent. In this way, the displacement function estimates the time taken to overcome an obstacle or go through a passageway on the study platform, based on the intrinsic characteristics of an individual (BMI, age, and seniority) from the reference contingent during the EER process.

Given the disparate data collected from the different tests conducted on various passageways or obstacles on the platform, the same steps were taken for each contingent and each tested element or obstacle. This ensures a range of displacement models that quantifies the time it takes for a member of any contingent to move along the established evacuation routes on the platform. Therefore, an SDE was carried out for each test of the reference contingents, resulting in two separate lines of data analysis (LF and VC), to which refinement and regression techniques were applied.

## Comparing tests between populations

After the data collected in the executed tests (longitudinal corridor and going through a hatch between decks and a watertight door) were cleaned and analyzed, regardless of the reference contingent, a displacement model was created for each test. Therefore, since the control groups are representative groups of their respective contingents (VC and LF), each contingent has a particular displacement model for each passage element or obstacle, modeled by regression functions that estimate the transit time of one of its members based on his/her particular characteristics (BMI, age, and seniority).

Even assuming that the tests conducted by each reference contingent were independent due to their distinct characteristics, a rigorous investigation required a new phase of data refinement and analysis supported by the following tests to accept or reject this assumption:

- Normal distribution test: a test that verifies if the data comes from a population where the variable follows a normal distribution with a mean and standard deviation equal to the one observed in the sample data.

- Independence test: a test that determines if two factors are independent and aims to verify if there is a dependent relationship between their data.

- Test of Homoscedasticity: a non-parametric test that compares the variance of a range of samples based on their median, assuming equal variances for normal populations as the null hypothesis of the test.

**Table 10. Results from various tests and VC SDE data.**

| Corridor (5 m) | Watertight door | Hatches | Tests with crew members Time |
|---|---|---|---|
| Yes ($p_{value}$ = 0.05) | Yes ($p_{value}$ = 0.23) | Yes ($p_{value}$ = 0.05) | Normality |
| Yes ($DW$ = 1.738) | Yes ($DW$ = 1.898) | Yes ($DW$ = 1.88) | Independence |
| No ($p_{value}$ = 0.136) | No ($p_{value}$ = 0.063) | Yes ($p_{value}$ = 0.012) | Heteroscedasticity |
| No ($p_{value}$ = 0.00) | No ($p_{value}$ = 0.04) | No ($p_{value}$ = 0.00) | ANOVA |

- ANOVA: a statistical procedure that checks if the variance is partitioned as a result of some explanatory variables. The technique is extensively used in the SDE to evaluate the effect of several treatments on the variability of the response variable, displaying significant differences between their results or accepting the null hypothesis that their population means do not differ.

Having listed the tests used for the data cleansing and analysis of the data collected in the trials with the control groups consisting of members of the crew of the study vessel (see Table 10), the following observations can be made:

- After applying the Ryan-Joiner normality test, the data collected from the three conducted trials does not allow us to state that there are sufficient indications to reject its null hypothesis, which is that these data follow a normal distribution.

- The conventional independence test was not considered because the groups of subjects involved in the same trial are homogeneous, invalidating the independence premise from the start due to the undeniable dependence between them. For these particular cases, the Durbin-Watson statistic is applicable. It confirms the independence of the data available from the three trials.

- The Levene test was performed to test homoscedasticity, which allows this comparison to be made regardless of the number of groups or samples. Thus, the premise of homoscedasticity between the experiments of the tests (longitudinal corridor and watertight door) is confirmed, while the last test (going through the hatch) is heteroscedastic.

- Finally, a Welch's ANOVA test was performed, which requires the validation of the following essential conditions:

  - The samples are validated to follow a normal distribution using the Shapiro-Wilk normality test, a test similar to the Ryan-Joiner test provided by the statistical package used.

  - The samples are independent of each other.

  - The samples are verified to have heteroscedastic variances according to the Levene test.

Based on these assumptions, the first two trials are immediately ruled out, and as a result of the ANOVA, the third trial is also discarded, thus rejecting the null hypothesis of equal means for the different trials.

According to these results and since the data from the different trials do not pass the ANOVA analysis, they do not validate their null hypothesis and, therefore, the available data do not allow us to assume that they have the same population mean, thus justifying the application of the independent displacement model for each trial constructed from the regression analysis using the SDE.

**Table 11. Results from various tests and LF SDE data.**

| Tests (SDE) | | | Tests with crew members Time |
|---|---|---|---|
| **Corridor (5 m)** | **Watertight door** | **Hatches** | |
| Yes ($p_{value}$ = 0.058) | Yes ($p_{value}$ = 0.05) | Yes ($p_{value}$ = 0.062) | Normality |
| Yes ($DW$ = 2.021) | Yes ($DW$ = 2.221) | Yes ($DW$ = 1.759) | Independence |
| Yes ($p_{value}$ = 0.03) | Yes ($p_{value}$ = 0.02) | Yes ($p_{value}$ = 0.01) | Heteroscedasticity |
| No ($p_{value}$ = 0.00) | No ($p_{value}$ = 0.02) | No ($p_{value}$ = 0.031) | ANOVA |

The same tests listed for the data cleansing and analysis phase of the trials with the control groups consisting of LF members were applied and the results (see Table 11) reveal the following:

- After applying the Ryan-Joiner normality test, the data collected from the three trials do not show that there are sufficient indications to reject the null hypothesis since these data follow a normal distribution.

- Conducting a conventional independence test is not feasible as the subjects form homogeneous groups involved in the same experiment, invalidating the independence premise due to the undeniable dependence between them. In such cases, the Durbin-Watson statistic, which confirms the independence of the available data in the three experiments, is applicable.

- The homoscedasticity test was performed using the Levene test, which allows for this comparison regardless of the number of groups or samples. Thus, the heteroscedasticity premise among the experiments of the trials (longitudinal corridor, hatch cover, and interdeck skylight) is confirmed.

- Finally, an ANOVA is performed based on the Welch test, which requires validation of the following premises:

  - The samples are validated to follow a normal distribution using the Shapiro-Wilk normality test, a test similar to the Ryan-Joiner test provided by the employed statistical package.

  - The samples are independent of each other.

  - The samples verify that their variances are heteroscedastic according to the Levene test.

As a result of these premises and the application of the ANOVA test, the null hypothesis of equal means for the different tests is rejected. Therefore, according to the results and since the tests do not have the same population mean, the development of a separate displacement model for each of them is justified, using the regression analysis process defined by the SDE.

Likewise, based on the results and given that the data from the different trials do not pass the ANOVA analysis, the null hypothesis is not validated, and, therefore, the available data do not allow us to assume that they have the same population mean. This supports the use of the independent displacement model for each trial, prepared from the regression analysis following the SDE.

In view of these results, the preparation of independent displacement models for reference passage or obstacle elements is justified. Furthermore, it is possible to verify whether the characteristics of the study contingents (VC vs. LF) or their respective levels of instruction and

**Table 12. Results from various tests, VC vs. LF.**

| Corridor (5 m) | Tests (SDE) Watertight door | Hatches | Tests Time (Crew members) |
|---|---|---|---|
| Yes ($p_{value} > 0.1$) | Yes ($p_{value} > 0.1$) | Yes ($p_{value} > 0.1$) | Normality |
| Yes ($DW = 1.738$) | Yes ($DW = 1.898$) | Yes ($DW = 1.88$) | Independence |
| **Corridor (5 m)** | **Watertight door** | **Hatches** | **Time (Landing forces)** |
| Yes ($p_{value} > 0.1$) | Yes ($p_{value} > 0.1$) | Yes ($p_{value} > 0.1$) | Normality |
| Yes ($DW = 2.021$) | Yes ($DW = 2.221$) | Yes ($DW = 1.759$) | Independence |
| **Corridor (5 m)** | **Watertight door** | **Hatches** | **Time (VC vs. LF)** |
| No ($p_{value} = 0.112$) | Yes ($p_{value} = 0.038$) | Yes ($p_{value} = 0.001$) | Heteroscedasticity |
| No ($p_{value} = 0.0$) | No ($p_{value} = 0.0$) | No ($p_{value} = 0.0$) | ANOVA |

training in the general layout of the vessel and emergency procedures are substantially different among their members.

Thus, the data captured in the different trials were compared aggregately and independently of the contingent to which their control groups belong (VC and LF), obtaining the results that substantiate the following statements (see Table 12):

- As discussed in previous sections, the data collected in the different tests with the control groups belonging to the study contingents passed the normality test.

- The Durbin-Watson statistic confirms the independence of the data collected in the trials with the control groups belonging to the study contingents.

- Levene's test for homoscedasticity was performed on the data from each of the trials, regardless of the group's affiliation, and it corroborated the homoscedasticity premise for the first trial (longitudinal corridor) and rejected it for the other two scenarios (watertight door and hatch between decks).

- Finally, in accordance with the previously stated conditions for the ANOVA analysis, the hypothesis of equal means for the first trial (longitudinal corridor) is rejected. After applying the ANOVA to the data from the remaining trials, it is not possible to validate the same null hypothesis.

Therefore, the data from the different trials do not pass the ANOVA test and do not validate the null hypothesis, which means that the available data do not allow us to assume that they have the same population mean. This explains the application of an independent displacement model for each group, which is composed of the regression functions constructed for each trial in accordance with the SDE.

## Summary of the results

Based on the characteristics considered by the SDE for an individual (BMI, age, and seniority) and regardless of his/her originating group, to quantify displacement when going through a passage or overcoming an obstacle on the study platform, a displacement model is required that considers the characteristics of the individual in question for each passage or obstacle.

In accordance with the differences observed in instruction and training of the members of the study contingents (VC vs. LF), their displacement models will also be different and, therefore, the characteristics considered for an individual will have different weights as well, acting as differentiating elements in the study contingents.

The differences observed by the ANOVA between the reference contingents, although not very significant in the corridor displacement test (around 10%), are indeed quite remarkable in the tests involving opening and closing watertight doors and hatches between decks, where the average times employed can vary up to thirty percent (30%) of the average time used.

In summary, given the characteristics of the study, each reference contingent has a different displacement model composed of functions applicable to each passage element or obstacle (longitudinal corridor, watertight door, and hatch), as can be expected due to their uniqueness.

The following figures show our conclusions regarding the differences among the members of the study contingents in going through the passage or overcoming obstacles on the reference platform. While the LF members are faster going through the corridor (Fig 5), the VC members are quicker going down the hatch and opening and closing the watertight door (Figs 6 and 7).

All these differences are verified by the ANOVA test, although different speeds are not noticeable in Fig 5 (displacement through corridor) with a difference of around ten percent, but they are significant in Figs 6 and 7 (overcoming watertight hatch and deck hatch), where the differences are around thirty percent of the average times taken.

Thus, the displacement models are not compatible and each contingent member will have to use his/her corresponding displacement model according to their contingent and the passage or obstacle in question.

## Conclusions

Data collected in the tests carried out (opening and closing a longitudinal corridor, overcoming a watertight door, and going down a hatch between decks) by the control groups of both reference contingents do not show equality in their population means. Thus, the application of

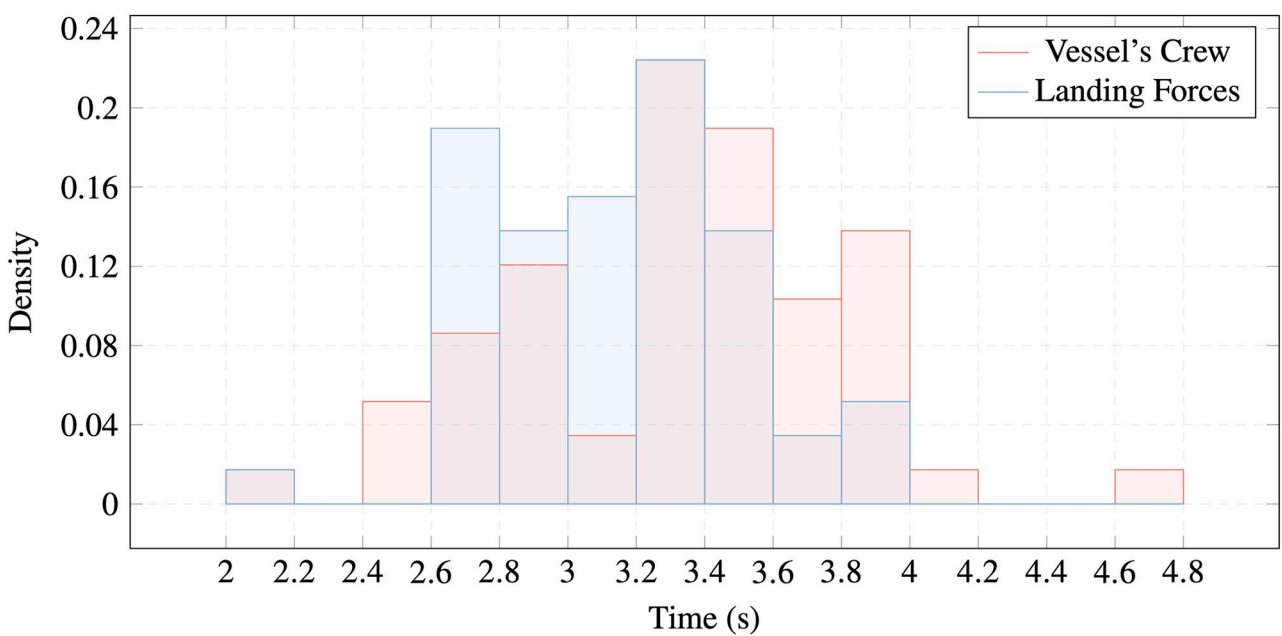

**Fig 5. Displacement times of contingents.** Comparison of corridor displacement times of VC vs. LF.

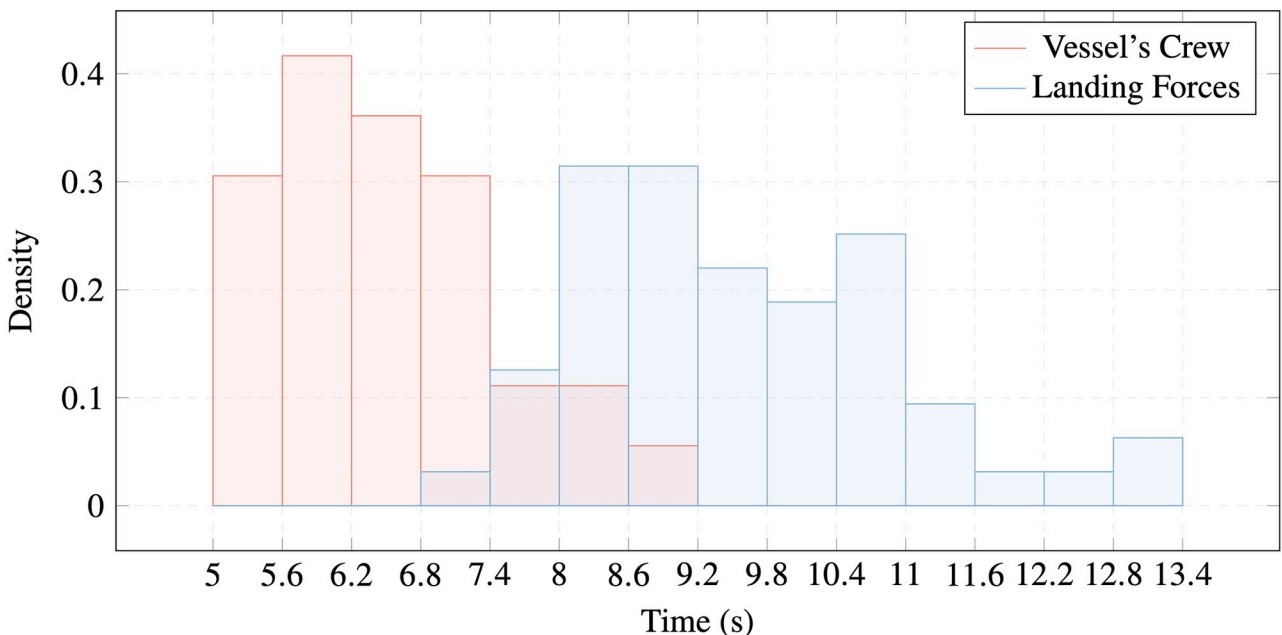

**Fig 6. Displacement times of contingents.** Comparison of opening and closing watertight doors times of VC vs. LF.

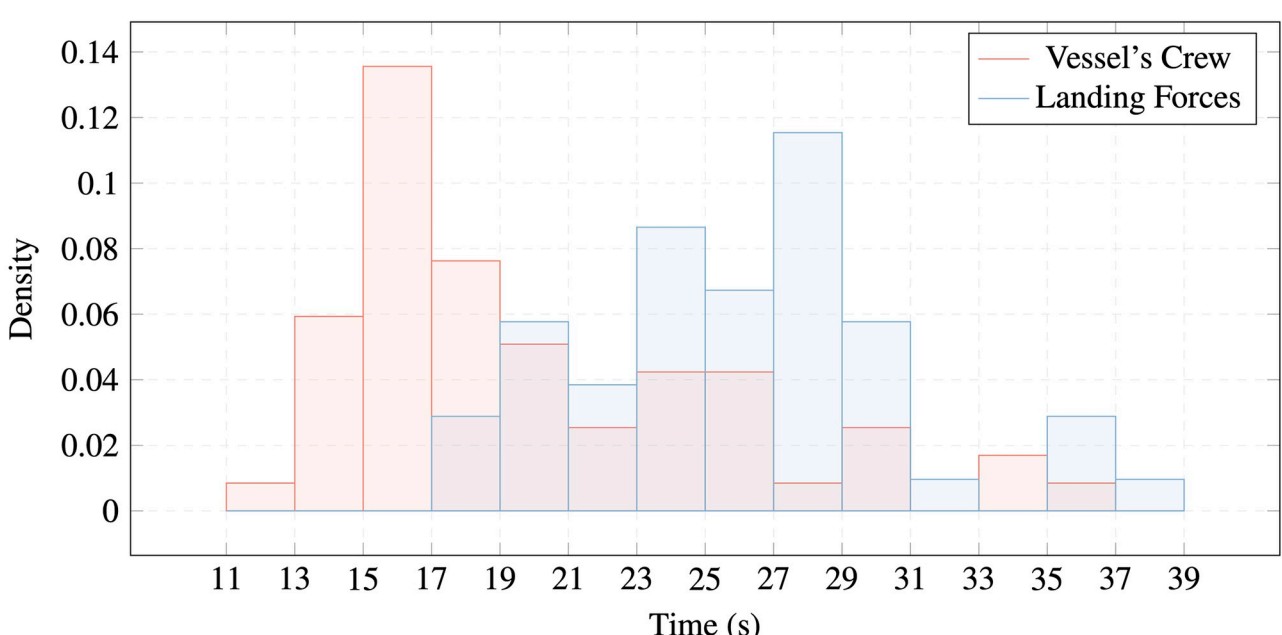

**Fig 7. Displacement times of contingents.** Comparison of hatch crossing between decks times of VC vs. LF.

the corresponding displacement function for the reference obstacle is justified to estimate the time taken by an individual with specific characteristics (BMI, age, and seniority) to overcome it.

1. Data collected in the various trials (displacement through a longitudinal corridor, opening and closing a watertight door, and going down a hatch between decks), grouped independently of the reference contingent, do not validate their equality of population means. Thus, the corresponding displacement function created for a trial's passage element or obstacle for an individual belonging to a particular contingent should be applied to estimate the time she/he takes to overcome it, according to their specific characteristics (BMI, age, and seniority).

2. The coefficients assigned to the factors or characteristics (BMI, age, and seniority) involved in the model will estimate the displacement of an individual through a defined passageway or obstacle and also act as differentiating elements for the displacement model of each reference contingency on the study platform.

Thus, the initial assumptions are confirmed, as the uniqueness of the study contingents results in significant differences depending on their level of knowledge about the general layout of the vessel and their training and instruction in the safety plans and procedures of the reference ship. All of this is reflected in their displacement model, which acts as a differentiating element in their ability to move during the execution of an EER process on board the study platform.

Considering the study's findings, it is recommended that relevant jurisdictions adopt policies that promote data-driven approaches to decision making. Thus, ship evacuation plans should take into consideration and adjust to the demographic characteristics of the embarked population. In particular, these authorities should mandate regular data collection and validation to ensure the accuracy and relevance of data sets used in policy planning and plan development.

## Supporting information

**S1 File.**
(PDF)

## Author Contributions

**Conceptualization:** Heitor Martinez-Grueira, Rafael Asorey-Cacheda.

**Data curation:** Joan Garcia-Haro.

**Formal analysis:** Heitor Martinez-Grueira.

**Funding acquisition:** Antonio-Javier Garcia-Sanchez.

**Investigation:** Heitor Martinez-Grueira.

**Methodology:** Joan Garcia-Haro.

**Project administration:** Joan Garcia-Haro.

**Resources:** Antonio-Javier Garcia-Sanchez.

**Software:** Heitor Martinez-Grueira.

**Supervision:** Rafael Asorey-Cacheda, Joan Garcia-Haro.

**Validation:** Rafael Asorey-Cacheda, Antonio-Javier Garcia-Sanchez, Joan Garcia-Haro.

**Writing – original draft:** Heitor Martinez-Grueira.

**Writing – review & editing:** Rafael Asorey-Cacheda.

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
