## [Decision Letter · Decision Letter 0]

12 Nov 2024

PONE-D-23-34444Understanding displacement of onboard contingents in Navy amphibious shipsPLOS ONE

Dear Dr. Asorey-Cacheda,

Thank you for submitting your manuscript to PLOS ONE. After careful consideration, we feel that it has merit but does not fully meet PLOS ONE’s publication criteria as it currently stands. Therefore, we invite you to submit a revised version of the manuscript that addresses the points raised during the review process.

We look forward to receiving your revised manuscript.

Kind regards,

Chun Kai Leung, Ph.D.

Academic Editor

PLOS ONE

Journal Requirements:

3. Thank you for stating the following financial disclosure: [This work was supported by the grants PID2020-116329GB-C22 and TED2021-129336B-I00 funded by MCIN/AEI/10.13039/501100011033 and by the European Union NextGenerationEU/PRTR.].

4. Please expand the acronym “AEI” (as indicated in your financial disclosure) so that it states the name of your funders in full. This information should be included in your cover letter; we will change the online submission form on your behalf.

5. Thank you for stating the following in the Acknowledgments Section of your manuscript: This work was supported by the grants PID2020-116329GB-C22 and TED2021-129336B-I00 funded by MCIN/AEI/10.13039/501100011033 and by the European Union NextGenerationEU/PRTR] We note that you have provided funding information that is not currently declared in your Funding Statement. However, funding information should not appear in the Acknowledgments section or other areas of your manuscript. We will only publish funding information present in the Funding Statement section of the online submission form. Please remove any funding-related text from the manuscript and let us know how you would like to update your Funding Statement. Currently, your Funding Statement reads as follows: [This work was supported by the grants PID2020-116329GB-C22 and TED2021-129336B-I00 funded by MCIN/AEI/10.13039/501100011033 and by the European Union NextGenerationEU/PRTR.] Please include your amended statements within your cover letter; we will change the online submission form on your behalf.

6. Thank you for stating the following in your Competing Interests section: [NO authors have competing interests]. Please complete your Competing Interests on the online submission form to state any Competing Interests. If you have no competing interests, please state "The authors have declared that no competing interests exist.", as detailed online in our guide for authors at http://journals.plos.org/plosone/s/submit-now This information should be included in your cover letter; we will change the online submission form on your behalf.

7. Please upload a copy of Supporting Information Figure/Table/etc. S1 Appendix. Normality tests which you refer to in your text on page 13.

Reviewers' comments:

Reviewer's Responses to Questions

**Comments to the Author**

1. Is the manuscript technically sound, and do the data support the conclusions?

Reviewer #1: Yes

Reviewer #2: Yes

2. Has the statistical analysis been performed appropriately and rigorously? 

Reviewer #1: Yes

Reviewer #2: Yes

3. Have the authors made all data underlying the findings in their manuscript fully available?

Reviewer #1: Yes

Reviewer #2: No

4. Is the manuscript presented in an intelligible fashion and written in standard English?

Reviewer #1: Yes

Reviewer #2: Yes

5. Review Comments to the Author

Reviewer #1: Excellent summary of a complex and important study. The supporting documentation and detailed method section with additional images and tables provide background to the project. There is nothing I would change, but more background and more literature review for those not as versed in this area may help.

Reviewer #2: The manuscript is well written and I have some minor suggestions before the manuscript is good to go.

1. Please make the underlying dataset available - this is PLOS ONE's policy unless there is an justification (e.g., privacy, national security information) for not releasing the dataset publicly.

2. Line 52, "Finally, Section “Conclusions” concludes", this phrase seems a bit weird. Can you rephrase it?

3. Have you got an ethical approval or at least an informed consent? PLOS ONE is very strict about this. Do mention it in your methodology or refer to the guidelines for authors.

4. Can you provide a descriptive statistics of your research participants' demographics , such as age, gender, BMI, or seniority, etc.

5. Can you be more specific about the policy implications? Perhaps you can suggest what policy should relevant legal authorities should enact in light of your results (One paragraph).

Overall a good job. If you cannot address some of the comments, please let me know why.

6. PLOS authors have the option to publish the peer review history of their article (what does this mean?). If published, this will include your full peer review and any attached files.

Reviewer #1: **Yes: **Bradley Wade Bishop

Reviewer #2: No

---

## [Author Response · Author response to Decision Letter 0]

21 Nov 2024

Response to editors:

3. Thank you for stating the following financial disclosure: [This work was supported by the grants PID2020-116329GB-C22 and TED2021-129336B-I00 funded by MCIN/AEI/10.13039/501100011033 and by the European Union NextGenerationEU/PRTR.].

Due to the time that has passed since the paper submission, the financial disclosure has changed. Additionally, as any of funder had any role, we have added "The funders had no role in study design, data collection and analysis, decision to publish, or preparation of the manuscript." to the financial information:

This work was a result of the ThinkInAzul and AgroAlNext programmes, funded by Ministerio de Ciencia, Innovación y Universidades (MICIU) with funding from European Union NextGenerationEU/PRTR-C17.I1 and by Fundación Séneca with funding from Comunidad Autónoma Región de Murcia (CARM). This work was also supported by the grants PID2023-148214OB-C21 and TED2021-129336B-I00, funded by MICIU/AEI/10.13039/501100011033 and by the European Union NextGenerationEU/PRTR. This work was also funded by Fundación Séneca (22236/PDC/23). This research was also contextualized to DAIMon, a cascade funding action deriving from the Horizon Europe project aerOS, funded by the European Commission under grant number 101069732. The funders had no role in study design, data collection and analysis, decision to publish, or preparation of the manuscript.

4. Please expand the acronym “AEI” (as indicated in your financial disclosure) so that it states the name of your funders in full. This information should be included in your cover letter; we will change the online submission form on your behalf.

AEI states for “Agencia Estatal de Investigación”. The reason of only using the acronym is because it is how it is stated by AEI

5. Thank you for stating the following in the Acknowledgments Section of your manuscript: This work was supported by the grants PID2020-116329GB-C22 and TED2021-129336B-I00 funded by MCIN/AEI/10.13039/501100011033 and by the European Union NextGenerationEU/PRTR] We note that you have provided funding information that is not currently declared in your Funding Statement. However, funding information should not appear in the Acknowledgments section or other areas of your manuscript. We will only publish funding information present in the Funding Statement section of the online submission form. Please remove any funding-related text from the manuscript and let us know how you would like to update your Funding Statement. Currently, your Funding Statement reads as follows: [This work was supported by the grants PID2020-116329GB-C22 and TED2021-129336B-I00 funded by MCIN/AEI/10.13039/501100011033 and by the European Union NextGenerationEU/PRTR.] Please include your amended statements within your cover letter; we will change the online submission form on your behalf.

This is the funding statement we want to be included:

This work was a result of the ThinkInAzul and AgroAlNext programmes, funded by Ministerio de Ciencia, Innovación y Universidades (MICIU) with funding from European Union NextGenerationEU/PRTR-C17.I1 and by Fundación Séneca with funding from Comunidad Autónoma Región de Murcia (CARM). This work was also supported by the grants PID2023-148214OB-C21 and TED2021-129336B-I00, funded by MICIU/AEI/10.13039/501100011033 and by the European Union NextGenerationEU/PRTR. This work was also funded by Fundación Séneca (22236/PDC/23). This research was also contextualized to DAIMon, a cascade funding action deriving from the Horizon Europe project aerOS, funded by the European Commission under grant number 101069732. The funders had no role in study design, data collection and analysis, decision to publish, or preparation of the manuscript.

6. Thank you for stating the following in your Competing Interests section: [NO authors have competing interests]. Please complete your Competing Interests on the online submission form to state any Competing Interests. If you have no competing interests, please state "The authors have declared that no competing interests exist.", as detailed online in our guide for authors at http://journals.plos.org/plosone/s/submit-now This information should be included in your cover letter; we will change the online submission form on your behalf.

Please, indicate the following in the online submission form:

NO authors have competing interests.

7. Please upload a copy of Supporting Information Figure/Table/etc. S1 Appendix. Normality tests which you refer to in your text on page 13.

The Appendix “Normality tests” does not use supporting information. This appendix is introduced to explain the calculation method for this type of test, which is one of those used for the study presented in this article.

The reference list has been checked. Regarding retracted paper, we have not found any in the reference list.

Response to reviewers:

3. Have the authors made all data underlying the findings in their manuscript fully available?

Reviewer #1: Yes

Reviewer #2: No

Unfortunately, we cannot disclose the data sets used for this study. Since the data used to carry out this displacement study have been extracted from a range of tests carried out by military personnel on board a Spanish Navy ship, these data are the property of the General Staff of the Navy (EMA) and, ultimately, of the Spanish State. It is also possible to use such data for the performance of a duly motivated and justified research, being mandatory to obtain an official authorization that enables the collection and processing of such data. This authorization will be issued by the owner of the data, in this particular case, by the EMA.

5. Review Comments to the Author

Reviewer #1: Excellent summary of a complex and important study. The supporting documentation and detailed method section with additional images and tables provide background to the project. There is nothing I would change, but more background and more literature review for those not as versed in this area may help.

Thanks for your comments. We do really appreciate them.

Reviewer #2: The manuscript is well written and I have some minor suggestions before the manuscript is good to go.

1. Please make the underlying dataset available - this is PLOS ONE's policy unless there is an justification (e.g., privacy, national security information) for not releasing the dataset publicly.

Since the data used to carry out this displacement study have been extracted from a range of tests carried out by military personnel on board a Spanish Navy ship, these data are the property of the General Staff of the Navy (EMA) and, ultimately, of the Spanish State. It is also possible to use such data for the performance of a duly motivated and justified research, being mandatory to obtain an official authorization that enables the collection and processing of such data. This authorization will be issued by the owner of the data, in this particular case, by the EMA.

2. Line 52, "Finally, Section “Conclusions” concludes", this phrase seems a bit weird. Can you rephrase it?

The phrase has been rewritten as: 

Finally, Section “Conclusions” provides the closing remarks.

3. Have you got an ethical approval or at least an informed consent? PLOS ONE is very strict about this. Do mention it in your methodology or refer to the guidelines for authors.

Yes, all study participants were military personnel and provided informed consent. Moreover, we have modified the fourth paragraph of Section “Material and methods”, line 70, page 2, as follows:

To conduct the range of tests, the Spanish Navy was asked to designate the amphibious assault ship “Galicia” (L-51) as the study vessel during the course of regular national maneuvers in the Gulf of Cadiz and all participating military personnel provided informed consent. This type of ship was used since the platform's and its personnel's characteristics needed to match the planned research. This would be challenging due to the vessel's dimensions, the size of the reference population, and the heterogeneity of the personnel on board.

4. Can you provide a descriptive statistics of your research participants' demographics , such as age, gender, BMI, or seniority, etc.

This information is provided throughout the manuscript. Tables 1 and 2 describe the relevant demographic characteristics of the control groups. Thus, 10 control groups with members belonging to the demographic groups described in Tables 1, 2, and 6 were used for the study. Tables 3, 4, 5, 7, 8 and 9 show the values obtained for each of these groups (y0 to y9), while each row represents an individual belonging to one of the demographic groups. In this sense, to facilitate the interpretation of these tables (3, 4, 5, 7, 8, 9), we have chosen to add a new column on the left with the ID corresponding to the demographic factors in tables 1, 2 and 6.

5. Can you be more specific about the policy implications? Perhaps you can suggest what policy should relevant legal authorities should enact in light of your results (One paragraph).

Thanks for the suggestion. We have added the following paragraph at the end of the Section “Conclusions” (page 14, line 363):

Considering the study's findings, it is recommended that relevant jurisdictions adopt policies that promote data-driven approaches to decision making. Thus, ship evacuation plans should take into consideration and adjust to the demographic characteristics of the embarked population. In particular, these authorities should mandate regular data collection and validation to ensure the accuracy and relevance of data sets used in policy planning and plan development.

Overall a good job. If you cannot address some of the comments, please let me know why.

Thanks for this comment. We think we have addressed all of your comments, but the one regarding data availability due to security constraints.

---

## [Editor Report · Decision Letter 1]

9 Dec 2024

Understanding displacement of onboard contingents in Navy amphibious ships

PONE-D-23-34444R1

Dear Dr. Asorey-Cacheda,

We’re pleased to inform you that your manuscript has been judged scientifically suitable for publication and will be formally accepted for publication once it meets all outstanding technical requirements.

Kind regards,

Chun Kai Leung, Ph.D.

Academic Editor

PLOS ONE
---

## [Editor Report · Acceptance letter]

18 Dec 2024

PONE-D-23-34444R1 

PLOS ONE

Dear Dr. Asorey-Cacheda, 

I'm pleased to inform you that your manuscript has been deemed suitable for publication in PLOS ONE. Congratulations! Your manuscript is now being handed over to our production team.

Kind regards, 

on behalf of

Dr. Chun Kai Leung 

Academic Editor

PLOS ONE